

# *In vivo* function of *Pgβglu-1* in the release of acetophenones in white spruce

Melissa H. Mageroy[1,2], Denis Lachance[3], Sharon Jancsik[1], Geneviève Parent[4,5], Armand Séguin[3], John Mackay[4,5] and Joerg Bohlmann[1,6,7]

[1] Michael Smith Laboratories, University of British Columbia, Vancouver, British Columbia, Canada
[2] Norwegian Institute of Bioeconomy Research, Aas, Norway
[3] Laurentian Forestry Centre, Natural Resources Canada, Quebec, Canada
[4] Department of Wood and Forest Sciences, Laval University, Quebec, Canada
[5] Department of Plant Sciences, University of Oxford, Oxford, United Kingdom
[6] Department of Forest and Conservation Sciences, University of British Columbia, Vancouver, British Columbia, Canada
[7] Department of Botany, University of British Columbia, Vancouver, British Columbia, Canada

## ABSTRACT

Eastern spruce budworm (*Choristoneura fumiferiana* Clemens) (ESBW) is a major forest pest which feeds on young shoots of white spruce (*Picea glauca*) and can cause landscape level economic and ecological losses. Release of acetophenone metabolites, piceol and pungenol, from their corresponding glycosides, picein and pungenin, can confer natural resistance of spruce to ESBW. A beta-glucosidase gene, *Pgβglu-1*, was recently discovered and the encoded enzyme was characterized *in vitro* to function in the release of the defensive acetophenone aglycons. Here we describe overexpression of *Pgβglu-1* in a white spruce genotype whose metabolome contains the glucosylated acetophenones, but no detectable amounts of the aglycons. Transgenic overexpression of *Pgβglu-1* resulted in release of the acetophenone aglycons *in planta*. This work provides *in vivo* evidence for the function of *Pgβglu-1*.

## INTRODUCTION

Eastern spruce budworm (*Choristoneura fumiferiana* Clemens) (ESBW) is considered the most detrimental pest of spruce and fir forests in eastern North America. While populations of ESBW usually persist at endemic levels, outbreaks can last for years incurring landscape level ecological changes and major economic losses (*Chang et al., 2012*; *MacLean, 2016*). With climate change, outbreaks of ESBW are predicted to increase in frequency and severity (*Hennigar et al., 2013*). Current forest management practices to control ESBW outbreak include costly aerial spraying of *Bacillus thuringiensis* (Bt) and the insect growth regulator Mimic® (*NRCAN, 2016*). The Bt *cry1Ab* gene has been successfully overexpressed in white spruce and shown to be effective against ESBW (*Lachance et al., 2007*); however, commercial deployment of transgenic trees is not permitted in Canada.

Recently, natural resistance to ESBW was discovered in white spruce (*Picea glauca*). Resistant genotypes accumulated the acetophenone aglycons piceol and pungenol as well as the corresponding glucosides picein and pungenin. Non-resistant genotypes only

Corresponding author
Joerg Bohlmann,
bohlmann@msl.ubc.ca

accumulated the acetophenone glucosides (*Delvas et al., 2011*). *Parent et al. (2017)* showed that the aglycons piceol and pungenol are the active defense compounds that contribute to resistance. We also showed that gene expression of *Pgβglu-1* was positively correlated with resistance, and in *in vitro* assays the encoded PgβGLU-1 enzyme cleaved the acetophenone glucosides, picein and pungenin, producing the biologically active aglycons (*Mageroy et al., 2015*). However, function of PgβGLU-1 has not yet been proven *in planta* and remained a critical but elusive part of the proof of function.

Here we report the successful overexpression of *Pgβglu-1* in a white spruce genotype Pg653. While wildtype Pg653 plants do not accumulate detectable amounts of the acetophenone aglycons, overexpression of *Pgβglu-1* resulted in the *in planta* formation of piceol and pungenol.

## METHODS

### Vector construction, *Agrobacterium* transformation and plant regeneration of *Pgβglu-1* overexpression white spruce

The full-length cDNA of *pgβglu-1* (GenBank KJ780719) or a modified green fluorescence protein (*gfp*) (Cambia) coding sequence were first cloned using the Gateway System (Invitrogen) into vector pMJM, containing the maize (*Zea mays*) ubiquitin promoter and the 35S terminator (*Levée et al., 2009*), then digested with *Sbf* I and sub-cloned into the binary vector pCAMBIA2300 (Figs. S1 and S2). The resulting constructs were transformed into *Agrobacterium tumefaciens* strain C58 pMP90 (*Hellens, Mullineaux & Klee, 2000*). Agrobacterium transformation of white spruce somatic embryonal masses (line Pg653) and subsequent selection and growth of transformants was performed as described by *Klimaszewska et al. (2001)*. Kanamycin resistance was used as the selection marker. Somatic embryo maturation, germination, acclimatization and transfer of somatic seedlings to soil were performed according to *Klimaszewska, Rutledge & Séguin (2004)*.

### LRE-qPCR of embryogenic tissue and somatic seedlings

Linear regression of efficiency (LRE) qPCR (*Rutledge, 2011*) was used to confirm transformation and to measure and compare absolute transcript abundance levels in both embryogenic tissue and somatic seedlings of 11 selected *Pgβglu-1* and 11 *gfp* transformed lines. RNA was isolated from up to 100 mg fresh weight of embryogenic tissue or from the pooled epicotyls of two 2-months old somatic seedlings using the RNeasy Plant mini kit (Qiagen) with on-column RNase-Free DNase (Qiagen) treatment. Primers were designed as previously described (*Foster et al., 2015*) with primers PgβGLU1-f—5′-GCCATAAGGGAGGGAGCAG; PgβGLU1-r—5′-CTCGCCCACTCAAAGCCGT or GFP-f—5′- GCCCGACAACCACTACCTGA; GFP-r—5′-GCGGTCACGAACTCCAGCAG used to analyze the *gβglu-1* and *gfp* lines respectively. cDNA synthesis, primer design, and PCR thermocycling conditions were conducted as described by *Foster et al. (2015)* with the exceptions that a two-step amplification protocol of 45 cycles was used with a 120 s annealing/elongation step at 65 °C. Gene expression was normalized using the two white spruce reference genes YLS8 and EF1α (*Rutledge et al., 2013*). Transcript abundance

quantification was performed using a Java program based on linear regression of efficiency previously described (*Rutledge, 2011*).

## Plant growth conditions

After growth on germination media for three months, somatic seedlings were planted into cones (Figs. 1C–1D) and maintained in a greenhouse under natural light and growth lights (16 h; 600 W HPS). Temperatures were set with a low of 19 °C and, within the limitations of a greenhouse that is not fully temperature controlled, to a high of 23.5 °C. Plants were allowed to grow for eight months and then placed at 4 °C with minimal light for two weeks to induce flushing. Plants were placed on the benchtop at 22 °C for one week to transition from the cold and then moved into a growth chamber with 16 h light at 22 °C and 8 h dark at 16 °C.

## RT-qPCR of plants grown for six months

Total RNA was isolated from needles of plants grown for six months using PureLink® Plant RNA Reagent (ThermoFisher, Waltham, MA, USA) using approximately 100 mg tissue according to manufacturer's instructions. RNA integrity and concentration was measured using Bioanalyzer 2100 RNA Nano chip assays (Agilent, Santa Clara, CA, USA) following the manufacturer's protocol. Equal RNA amounts were used for cDNA synthesis with the iScript Reverse Transcription Supermix (Bio-Rad, Hercules, CA, USA). qRT-PCR reactions were performed on a Bio-Rad CFX96 Real-time system using the SsoFast kit (Bio-Rad, Hercules, CA, USA) in triplicate. Relative transcript abundance was calculated using efficiency corrected $\Delta C_T$ and $\Delta\Delta C_T$ values based on ELF-1α as the reference gene. Target-specific oligonucleotides were as follows: ELF-1α-f—5′-CCCTTCCTCACTCCAACTGCATA; ELF-1α-r—5′-TCGGCGGTGGCAGAGTTTACATTA; or PgβGLU1-f—5′-TTGGATCCTCTGAAGGT GT; PgβGLU1-r—5′-TCCCTCCCTTATGGCTTC. Target specificity was confirmed by sequence verification of representative amplicons.

## Metabolite analysis

For the time course study of acetophenone glucoside deglycosylation, tissue was ground and left on the bench top for 4 h, 8 h, and 24 h before adding extraction solvent (100% methanol containing 1 mg/ml benzoic acid as the internal standard). For all other metabolite extractions, 100 mg of tissue was placed in a vial and 1 mL of extraction solvent was immediately added. The vial was capped and placed at 4 °C; with shaking overnight. The supernatant was removed and placed in a new vial. For liquid chromatography-mass spectrometry (LC-MS) analysis, samples were diluted 1:10 by diluting 100 µL of supernatant in 900 µL of 100% methanol. LC-MS analysis was performed using a LC-MSD-Trap-XCT_plus with a SB-C18, 15-cm column (Agilent, Santa Clara, CA, USA). An injection volume of 10 µL was used. Solvent A was water with 0.2% (v/v) formic acid; solvent B was 100% (v/v) acetonitrile with 0.2% (v/v) formic acid. The following gradient was used: increase to 5% solvent B from 0 to 0.5 min; increase to 22% solvent B from 0.5 to 5.0 min; increase to 35% solvent B from 5.0 to 10.0 min; increase to 50% solvent B from 10.0 to 13.0 min; increase to 95% solvent B from 13.0 to 16.0 min; holding 95% solvent B

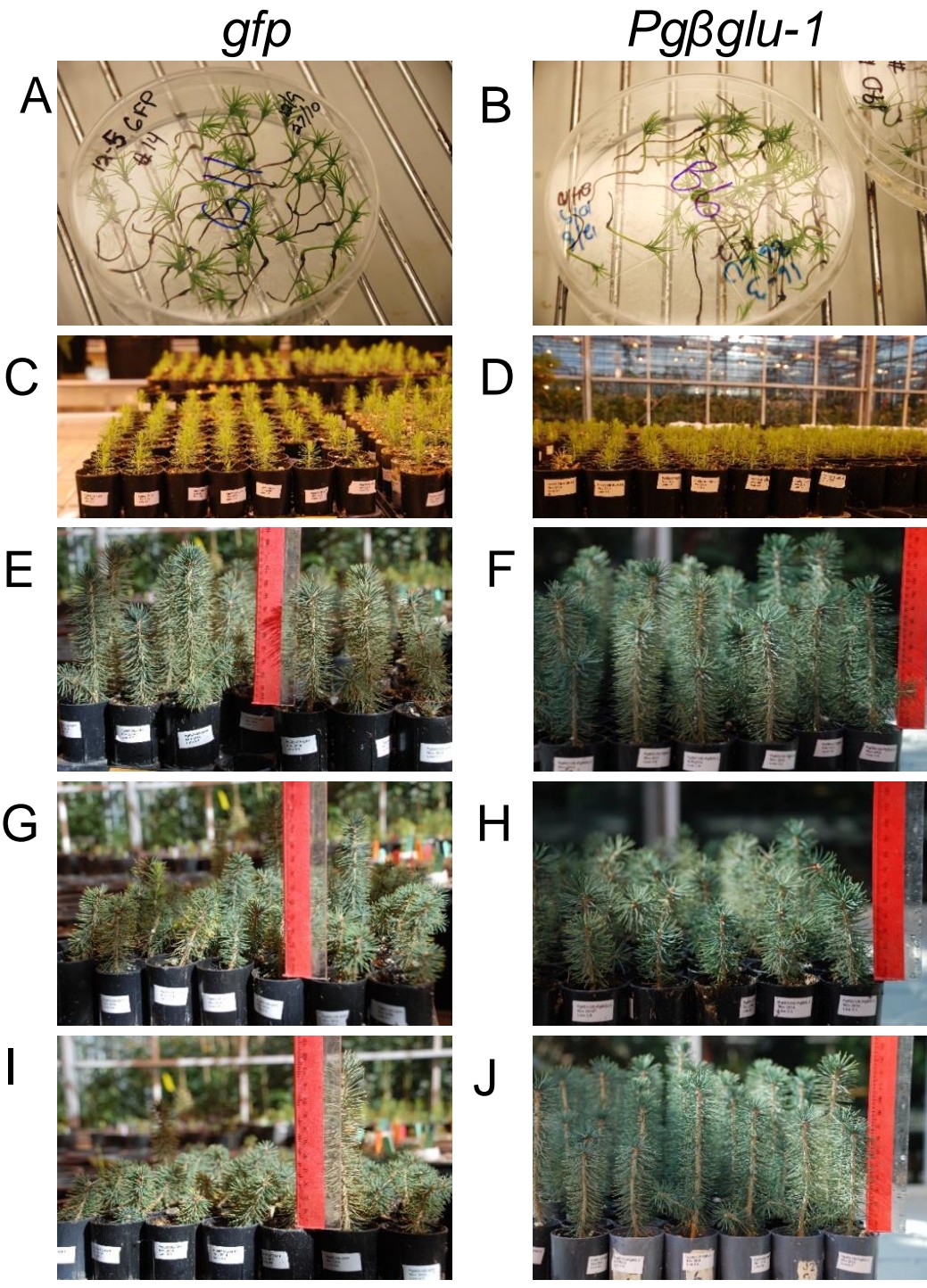

*gfp*    *Pgβglu-1*

**Figure 1 Transgenic white spruce seedlings.** (A), (C), (E), (G), and (I) show control white spruce seedlings expressing *gfp*. (B), (D), (F), (H), and (J) show white spruce seedlings overexpressing *Pgβglu-1*. (A–B) Transgenic white spruce somatic seedlings were grown on selective media for three months. (C–D) Plantlets were then transferred into cones and placed in the greenhouse. (E–J) After six months of growth, on average, the *Pgβglu-1* overexpressing seedlings appeared healthier and taller.

from 16.0 to 17.0 min; decrease to 5% solvent B from 17.0 to 17.1 min. Column flow rate was 0.8 mL min$^{-1}$. Piceol and picein were identified using the extracted ion 135(−), the parent mass (−1) of piceol. Pungenol and pungenin were identified using the extracted ion 151(−), the parent mass (−1) of pungenol.

## RESULTS AND DISCUSSION

### Overexpression of SBW defense gene *Pgβglu-1*

We overexpressed the cDNA of *Pgβglu-1 in planta* to validate the function of this gene and its encoded enzyme activity in the release of acetophenone aglycons from the corresponding glucosides in white spruce foliage (Fig. 1). We used the white spruce genotype Pg653 to test the effect of overexpression of *Pgβglu-1* for two reasons: (1) Pg653 is a well-established somatic embryogenic line for white spruce transformations. (2) This line shows a metabolite phenotype that contains the acetophenone glucosides picein and pungenin, which are the proposed *in vivo* substrates for Pgβglu-1 enzyme activity, but contains minimal detectable amounts of the corresponding aglycons piceol and pungenol. Thus, Pg653 provides a suitable background for *de novo* formation of piceol and pungenol in transgenic plants. *Agrobacterium* transformation of the coding region of *Pgβglu-1* driven under maize (*Zea mays*) ubiquitin promoter was used to produce transgenic white spruce lines. A *gfp* reporter gene was overexpressed in white spruce as a control.

### Evaluation of possible negative effects of transgene overexpression

Levels of transgene gene expression were evaluated in both embryonal tissue and somatic seedlings using qPCR (Fig. 2). Overall transcript levels of the *gfp* transgene were higher in both sample types compared to the *Pgβglu-1* transgene. The lower levels of *Pgβglu-1* transcripts could indicate some phytotoxic effects, as toxic compounds are often glycosylated in plants for self-protection. However, *Pgβglu-1* overexpressing young plants appeared to be healthier than *gfp* expressing plants (Figs. 1E–1J) under greenhouse conditions, including naturally occurring biotic and abiotic stresses. Although previous studies have shown *gfp* to be non-toxic in plants (*Millwood, Moon & Stewart Jr, 2010*; *Tian et al., 1999*), deleterious effects have been noted in mammalian cells (*Liu et al., 1999*). The observed plant growth difference may be due to physiological adaptations to cope with effects that may arise from high levels of *gfp* expression (*Steward, 2001*). For example, under stress conditions, which increase the production of free radicals, plant cells may not be able to compensate as well for high expression of *gfp* leading to negative growth effects.

### Young seedlings overexpressing *Pgβglu-1* do not accumulate aceotphenone aglycons

No significant levels of acetophenone aglycons were observed in either *gfp* controls or *Pgβglu-1* overexpressing transgenic seedlings after the first six months of growing in the greenhouse. To test if we could observe aglycon production in *Pgβglu-1* overexpressing foliage when tissues were disrupted, we chose one high, one medium, and one low *Pgβglu-1* expressing line based on transcript abundance in six-month old seedlings (Fig. 3A). Tissue was ground and the disrupted tissue left at room temperature for up to 24 h before
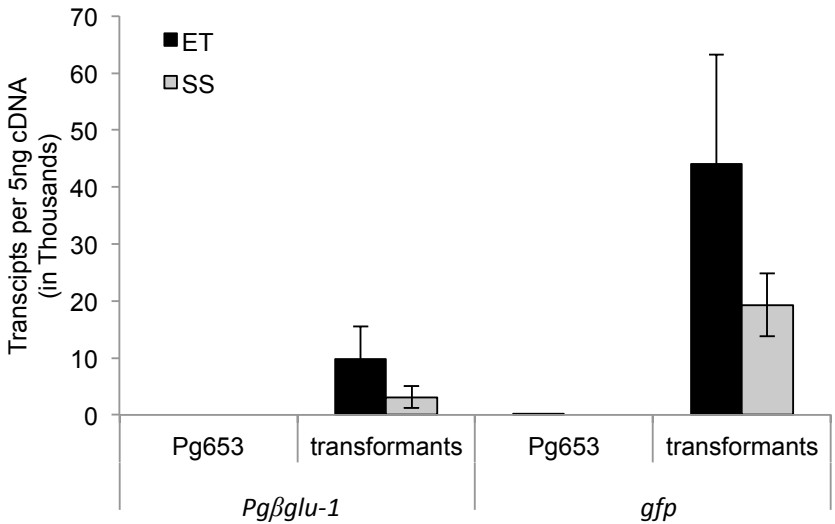

**Figure 2** **LRE-qPCR quantification of transgene expression in embryonic tissue (ET) and somatic seedlings (SS) of transformed lines.** Transcript abundance was calculated as the number of transcripts in 5 ng of synthesized cDNA. Pg653 represent the untransformed control line. Overall, higher expression of *gfp* was observed in ET and SS than *Pgβglu-1*. Error bars represent standard deviation. *N* = 11.

metabolite extraction. In this time course test, we observed much greater release of the acetophenone aglycons piceol and pungenol in *Pgβglu-1* overexpressing lines compared to *gfp* controls (Fig. 3B). In nature, acetophenone aglycons are produced in resistant white spruce foliage without tissue disturbance (*Mageroy et al., 2015*). The requirement of tissue disruption to produce the aglycons in the *pgβglu-1* overexpressing lines may indicate that PgβGLU-1 protein is prevented from interacting with glucosides in the young seedlings, perhaps due to differential localization of the enzyme and the substrate or some reversible inhibition or inactivation of the enzyme. As *Pgβglu-1* was expressed under a constitutive promoter it is plausible that this expression may be spatially and temporally amiss or that reversible protein modification rendered it inactive in young seedlings. However, the ability of *Pgβglu-1* overexpressing lines to produce greater amounts of aglycon when tissue was disturbed provided additional proof for the function of this gene and its encoded protein. It is possible that this increased accumulation of acetophenone aglycons upon tissue distruption may contribute enhanced resistance in plants overexpressing *Pgβglu-1*. This remains to be tested in future work.

## Seedlings overexpressing *Pgβglu-1* accumulate acetophenone aglycons in newly growing shoot tissue after induced dormancy

Since white spruce is a perennial tree species, acetophenone production may be influenced by plant development beyond the first growth phase. We tested this possibility by carrying eight-month old seedlings through a simulated complete growth cycle including bud set, winter dormancy, and new bud flush, which involved a cold treatment in the dark and subsequent return to normal light and temperature conditions favorable to active vegetative growth. Bud flush began three weeks after returning trees to normal growth

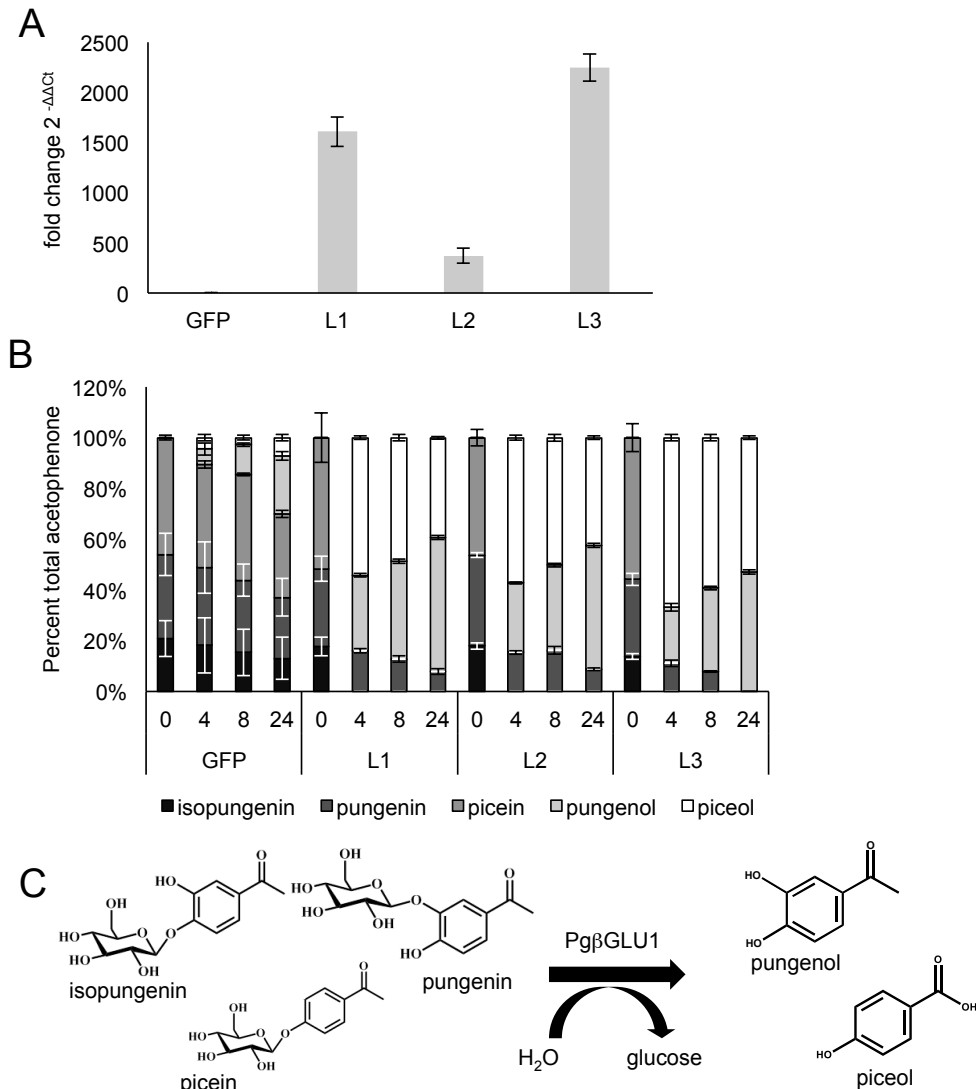

**Figure 3 Analysis of transgene expression and β-glucosidase potential in 6-months old transgenic white spruce seedlings.** (A) After six months of growth in the greenhouse, qRT-PCR was used to determine the fold change in *Pgβglu-1* expression between *gfp* expressing plants and three independent lines of *Pgβglu-1* overexpressing plants. Representative low, medium and high *Pgβglu-1* expressing lines were chosen for comparative metabolite analysis. GFP represents the average of four independent lines. Error bars represent standard deviation. $N = 3$. (B) To test if acetophenone glucoside could be released in *Pgβglu-1* overexpressing seedlings, needles were grounds and the disrupted tissue left for 0 h, 4 h, 8 h, and 24 h before extracting metabolites. A much larger proportion of acetophenone aglycons was released in *Pgβglu-1* overexpressing trees compared very small proportion of acetophenone aglycons released in *gfp* overexpressing tissue. Error bars represent standard error. $N = 3$. (C) The structures of glucosylated acetophenone and their aglycons with the catalytic function of PgβGLU-1.

conditions. Following a gap period of no detectable levels of acetophenone glucosides and aglycons in the newly flushing shoots, accumulation of both acetophenone glucosides and the corresponding aglycons was detected at eight weeks after the beginning of new shoot growth (Figs. 4A–4B). While both the acetophenone glucosides and aglycons were

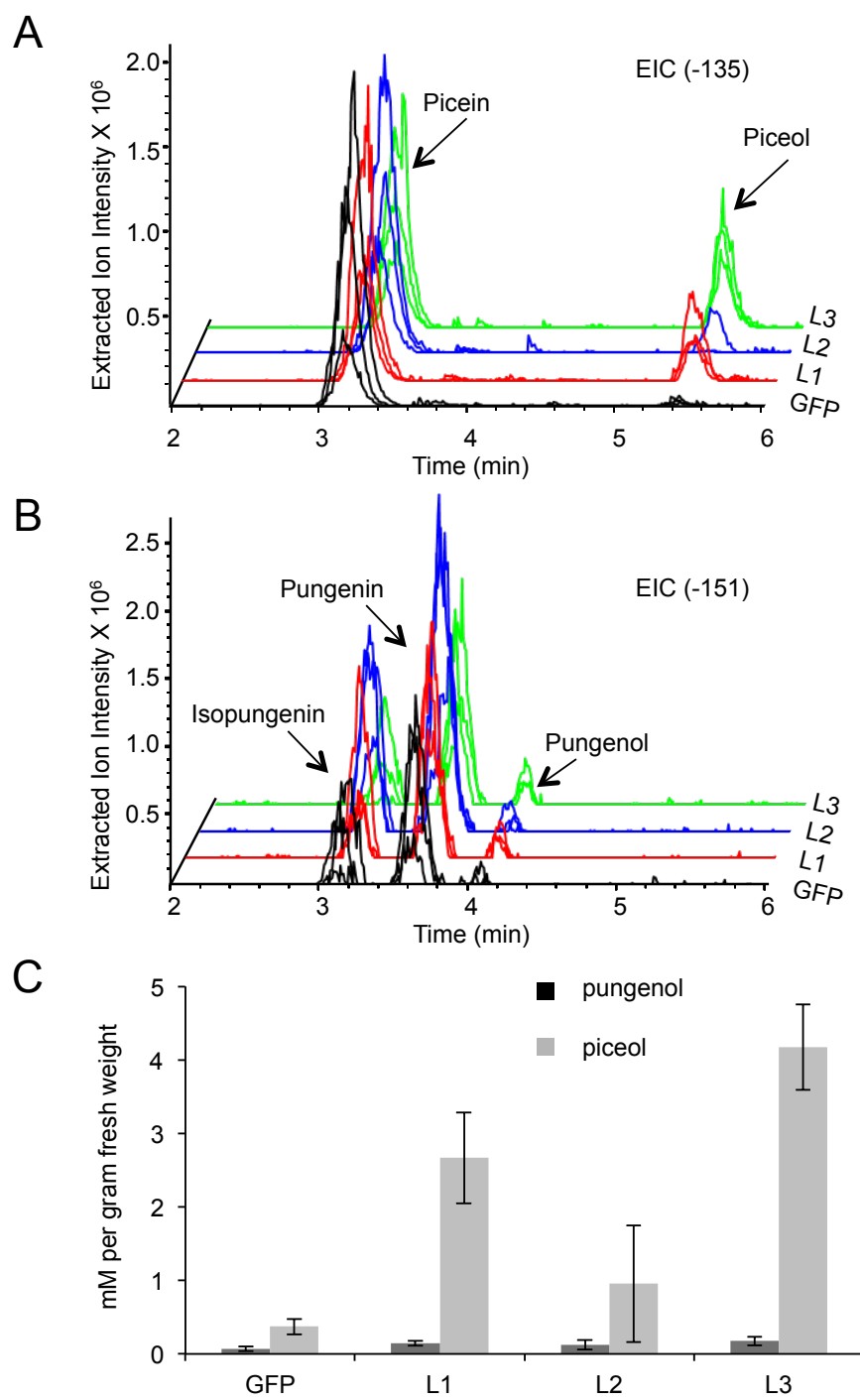

**Figure 4** **Altered acetophenone glucoside and aglycon profiles in new shoots of white spruce seedlings after bud flush.** Eight weeks after the beginning of bud flush, acetophenone aglycons were detected in extraction from intact *Pgβglu-1* overexpressing shoots. (A) The extracted ion chromatogram (EIC) for the parent mass of piceol ($-135$). (B) The EIC for the parent mass of pungenol ($-151$). (C) Piceol and pungenol were quantified using authentic standards. A higher amount of piceol was released in *Pgβglu-1* overexpressing shoots compared to the amount of pungenol. Error bars represent standard error. $N = 3$.

observed in the *Pgβglu-1* overexpressing seedlings, no substantial quantities of the aglycons were detected in the *gfp* transgenic control seedlings. Under these *in planta* conditions, overexpression of *Pgβglu-1* led to higher amounts of the picein-derived aglycon piceol compared to the pungenol aglycon (Fig. 4B). Accumulation of piceol also correlated with the difference of *Pgβglu-1* transcript levels in low, medium and high expressing lines (Figs. 3A and 4B). These results conclusively confirm *in planta* function of *Pgβglu-1* in the release of acetophenone aglycons and their accumulation in intact plant tissue. In previous work, we reported the *in vitro* kinetic parameters of the PgβGLU-1 enzyme with picein as the substrate, but not for pungenin as this substrate is not a commercially available (*Mageroy et al., 2015*). The present results suggest that PgβGLU-1 is more active on picein, compared to pungenin, *in planta*.

## CONCLUSIONS

We showed that overexpressing *Pgβglu-1* in a white spruce genotype that does not naturally contain acetophenone aglycons leads to the *in planta* formation of the resistance metabolite piceol, and in disrupted tissues also the additional formation of pungenol. The results validate previously reported *in vitro* function of *Pgβglu-1* and its encoded PgβGLU-1 enzyme. The different results obtained with young seedling before bud flush and seedlings that had passed through bud set and new bud flush point out the need for caution when evaluating phenotypes of young seedling overexpressing a transgene. As we found, the altered metabolite phenotype was not observable in intact tissue until after the first bud flush. Conditions of spruce metabolism that provide the precursors for altered metabolism may vary depending on the developmental stage of seedlings, where precursors for defense metabolism may only become fully accessible after the seedlings have gone through an initial growth phase or a dormancy phase. In future work, effects of the *Pgβglu-1* transgene expression and altered acetophenone profiles in transgenic Pg653 trees will be tested with insect feeding test, which will require production and maturation of a larger number of young trees.

## ACKNOWLEDGEMENTS

We thank Lina Madilao for her assistance with LCMS. JB is a UBC Distinguished University Scholar.

### Funding

The work was supported by the Natural Sciences and Engineering Research Council of Canada (NSERC; Strategic Project Grant to JB and JM and Discovery Grant to JB); the Genomics R&D Strategy of Canada (to AS); and funds received through Genome Canada, Genome British Columbia and Genome Quebec for the SMarTForests Project (to JB and JM) and the Spruce-Up (243FOR) Project (to JB). The funders had no role in study design, data collection and analysis, decision to publish, or preparation of the manuscript.

## Grant Disclosures

The following grant information was disclosed by the authors:
Natural Sciences and Engineering Research Council of Canada.
Genomics R&D Strategy of Canada.

## Competing Interests

The authors declare there are no competing interests.

## Author Contributions

- Melissa H. Mageroy conceived and designed the experiments, performed the experiments, analyzed the data, wrote the paper, prepared figures and/or tables, reviewed drafts of the paper.
- Denis Lachance conceived and designed the experiments, performed the experiments, analyzed the data, contributed reagents/materials/analysis tools, prepared figures and/or tables, reviewed drafts of the paper.
- Sharon Jancsik and Geneviève Parent performed the experiments, reviewed drafts of the paper.
- Armand Séguin conceived and designed the experiments, contributed reagents/materials/analysis tools, reviewed drafts of the paper.
- John Mackay reviewed drafts of the paper.
- Joerg Bohlmann conceived and designed the experiments, wrote the paper, reviewed drafts of the paper.

## Supplemental Information

Supplemental information for this article can be found online at http://dx.doi.org/10.7717/peerj.3535#supplemental-information.

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
