# Peer review of "In vivo function of Pgβglu-1 in the release of acetophenones in white spruce"

_PeerJ, doi:10.7717/peerj.3535_

## Round 0.1 · original submission · Minor Revisions

Overall the reviews were very positive. Some small amendments have been suggested. Please, note the annotated manuscript in the attachment.

·

Basic reporting

This was one of the easiest manuscripts I have had to review in recent memory. Very straightforward, well clearly written, rigorously conducted.

Experimental design

Everything appears to be very solid to me and all requirements are met.

Validity of the findings

Discussion and conclusion are concise and to the point, clear and well supported by the data.

Additional comments

This is a very good paper. I have nothing substantial to recommend in terms of changes. I would only suggest the authors briefly discuss the meaning of the accumulation of aglycones in ground tissue in relation to what happens when the insect chews the needles (a passive but perhaps significant response to feeding). Constitutive levels are important, but so are those related to mechanical damage.

I caught a few typos which I noted in the attached PDF.

Reviewer 2 ·

Basic reporting

no comments

Experimental design

no comments

Validity of the findings

no comments

Additional comments

The manuscript shows very interesting findings and deserve to be published

Reviewer 3 ·

Basic reporting

This is a good follow-up report of Delvas et (2011) and Mageroy et al (2015) to show the impact of overexpressing beta-glucosidase gene in spruce cultivar that does not accumulate aglycons. The objective of the project is clearly stated in the introduction, and transformation of spruce was performed to achieve the objective. As spruce transformation is not trivial, this paper includes substantial amount of work. Obviously, this work can be more complete and significant if the possible defensive phenotype of the transgenic spruce is tested with ESBW, but such ecological data is beyond the scope of this work I guess. Confirmatory data of the transgenic spruce using qPCR and LC-MS analyses were solid, and authors delivered well that the multiple spruce lines were transformed. Although authors acquired negative data at the beginning, but they meticulously pursue the changes in chemical phenotype in the second year to show the transgenic spruces uniquely synthesize aglycons.

Although speculative, authors provide explanations as to why the aglycons could be only detected in newly growing shoot tissue after dormancy. More work is necessary to understand better about the physiology of perennial plant.

Overall, it is a concise and well-written article that includes new data in tree biology.

Experimental design

Research question is well defined a the beginning, and following experiments were very focused to address the questions. However, I have two minor comments

The qPCR data in 3A is based in delta-delta Ct value, and its methodology was given in the method. However, sufficient method was not given for the qPCR-data in Figure 2. The Y-axis seems to indicate absolute abundance of the transcript number; however, it is not clear how authors obtained such data. More methodological information is required.

Figure 1 legend and respective text - "...appeared healthier and tall" read very subjective and non-scientific. Can authors define "healthier and tall" in a more objective manner.

Validity of the findings

Ranges of the data variations can be provided for Figure 3b. Error bars can be shown, but if the presentation of the error bars may complicate the figure presentation, authors simply state a sentence such as that "for all samples, less than xx% STD or SE was calculated".

Additional comments

Structures for isopungenin, pungenin, picein, and piceol can be given in Figure 3 or in a separate Figure (possibly Figure 1), together with the catalytic function of the enzyme in the figure.

---

## Round 0.2 · accepted · Accept

Congratulations on the acceptance of your manuscript!

Reviewer 3 ·

Basic reporting

Authors addressed all questions raised by this reviewer and others. This manuscript is suitable for publication in PeerJ.

Experimental design

More detailed qPCR methodologies were added.

Validity of the findings

Fine

Additional comments

All questions were properly addressed. Thank you for your effort to improve the manuscript.

---

## Author Rebuttal · Round 0.2

We thank the Editor and the two reviewers (1 and 3) for their very positive assessment of our paper. The review comments provided suggestions for minor editorial revisions, which have been made in the revised paper as described below. In addition we made a few minor spelling and formatting corrections, which are shown in the submitted track-change version of the revised submission.

**Response to Reviewer 1**

**Comment**: I would only suggest the authors briefly discuss the meaning of the accumulation of aglycones in ground tissue in relation to what happens when the insect chews the needles (a passive but perhaps significant response to feeding). Constitutive levels are important, but so are those related to mechanical damage.

**Response**: We have added wording in the discussion about the possibility of increased accumulation of aglycones in disrupted tissue contributing potentially to enhanced resistance in plants overexpressing *Pgβglu-1* (lines 162 -165).

**Response to Reviewer 3**

**Comment**: The qPCR data in 3A is based in delta-delta Ct value, and its methodology was given in the method. However, sufficient method was not given for the qPCR-data in Figure 2. The Y-axis seems to indicate absolute abundance of the transcript number; however, it is not clear how authors obtained such data. More methodological information is required.

**Response**: Requested additional details have been included to the methods section (lines 70-72).

**Comment**: Figure 1 legend and respective text - "...appeared healthier and tall" read very subjective and non-scientific. Can authors define "healthier and tall" in a more objective manner.

**Response**: We rephrased the wording in the legend of Figure 1 to "… **were taller and appeared more vigorous compared to the seedlings expressing** *gfp*." Although we did not measure other parameters, such as for example chlorophyll content, to further substantiate this observation, Figure 1C provides images of the seedlings to support this statement.

**Comment**: Ranges of the data variations can be provided for Figure 3b. Error bars can be shown, but if the presentation of the error bars may complicate the figure presentation, authors simply state a sentence such as that "for all samples, less than xx% STD or SE was calculated".

**Response**: Error bars have been added as requested.

**Comment**: Structures for isopungenin, pungenin, picein, and piceol can be given in Figure 3 or in a separate Figure (possibly Figure 1), together with the catalytic function of the enzyme in the figure.

**Response**: As requested, structures and a reaction scheme have been added to Figure 3.